# The Design and User Evaluation of Body-Transfer System via Sliding Transfer Approach for Assisting Functionally Impaired People

Chyi-Yeu Lin [1,2,3], Salman Masroor [1,*], Bahrudin [1] and Hasan Bulut [1]

1   Department of Mechanical Engineering, National Taiwan University of Science and Technology,
    Taipei 106, Taiwan; jerrylin@mail.ntust.edu.tw (C.-Y.L.)
2   Center for Cyber-Physical Systems, National Taiwan University of Science and Technology, Taipei 106, Taiwan
3   Taiwan Building Technology Center, National Taiwan University of Science and Technology,
    Taipei 106, Taiwan
*   Correspondence: d10603826@mail.ntust.edu.tw; Tel.: +886-2-2733-3141 (ext. 3290); Fax: +886-2-2730-1187

**Abstract:** Assistive devices can significantly improve caregivers' ability to help disabled people with their daily activities. Existing assistive devices are not fully capable of safe transfer and are still in their early stages of development. In this research, a body-transfer system is designed and developed to ensure that the posture and body angle of the person in the sagittal plane remains unaltered while transferring from bed to wheelchair and vice versa. Two independently controlled conveyor belts (2-DOF) mounted on the indigenously developed bed are employed to transfer the disabled person using a sliding approach. Additionally, a wheelchair with conveyor belts that are fully automated is used to carry and transfer the user to and from the wheelchair. Furthermore, an integrated control architecture has been developed for safely operating the entire body-transfer system (from an indigenously developed bed and wheelchair). Finally, an experimental assessment of the body-transfer system's performance has been conducted. The experimental findings demonstrate that the system can transfer up to 120 kg of body weight while the user's posture remains unaltered in the sagittal plane. Users perceive a reduction in wrist and shoulder pain index using the body-transfer system. The system has great potential for relocating disabled persons safely while reducing the load on caregivers.

**Keywords:** smart home; body-transfer bed; body-transfer bed wheelchair; toilet/bathroom patient transfer; transfer assistive system

## 1. Introduction

Strides made in the progress of medical science for increasing the average life expectancy of the global population have led to a gradual increase in citizens of senior age in society. However, with increasing age comes increasing immobility. Statistically, around 15% of the global population is beset by some mobility limitation, such as paralysis of the lower limbs [1]. China and India have approximately 24.12 and 5.4 million physically disabled people in their total populations, respectively, ranked the highest in Asia [2,3]. On the other hand, Japan and Taiwan are facing the challenge of becoming super-aged societies by 2061, with aged people already representing approximately 37.7% and 14.7% of the total population [4–7].

Elderly and physically disabled people usually depend on healthcare workers to assist them in performing activities of daily living (ADLs), such as moving to and from the bed to the wheelchair. However, the traditional transfer of such people to and from bed to wheelchair and performing toileting and bathing is highly challenging and inconvenient for disabled people and healthcare workers. Manual transfer accelerates injury likelihood for functionally disabled people and caretakers if performed incorrectly. Additionally, 55%

of healthcare professionals typically experience musculoskeletal disorders (MSDs), such as low-back pain (LBP), after working for 1 year [8]. Furthermore, compared to other categories of the working population, the stated prevalence is higher among healthcare professionals [9–11]. Thus, the demand for state-of-the-art patient transfer systems that are safe, human-friendly, and alleviate the assistance of healthcare workers is on the rise.

Generally, transferring from bed to wheelchair and performing bathing and toileting is a complex and challenging process. Many assistive types of machinery are needed to transfer a disabled person from bed to wheelchair and vice versa. To mitigate the complexity of assistance, Panasonic developed a hybrid wheelchair/bed with assist capabilities, fusing an electric care bed and a fully reclining wheelchair known as Panasonic Resyone [12,13]. A linear actuator is employed for the reclining operation of the wheelchair unit; similarly, three linear actuators are used to assist the bed unit's backrest, leg rest, and height. This makes the shifting process effortless. With the help of manual operation, the bed and wheelchair units can be combined to form a single unit. Research shows that most caretakers are ill-prepared for the assistant role and do not offer a high level of care, which is a sign of risk for the patient [14]. Traditional approaches are unable to offer risk-free assistance to both the person being cared for and the caregiver. On the other hand, proper and complex caregiving is stressful and can affect the psychological health of the caregiver.

This research aims to investigate the transfer of a person from a bed to a wheelchair and support with bathing and toileting issues using a sliding transfer approach. The principal difficulties that come up in the research are as follows:

1. How can a person be moved from a bed to a wheelchair without altering their posture and angle in the sagittal plane?
2. How can a physically disabled person attain active assistive living (AAL) using various smart home robotic modules?
3. How is a human-friendly and intelligent interface developed for a physically disabled person by which activities of daily living can be performed independently?

The main objectives of our proposed methodology are summarized as follows:

1. This research focuses on achieving AAL by utilizing a fully integrated body-transfer system to support the user in transferring from bed to wheelchair.
2. To mitigate the risk of injuries from a traditional lifting arrangement during the body transfer from a bed to a wheelchair, this is replaced with the sliding transfer approach. Furthermore, the developed system can assist users in the bathroom and toilet.
3. Furthermore, this study proposes a user-friendly architecture to efficiently manage all the body-transfer smart home assistive devices.

The sections of this paper are structured as follows: Initially, in Section 2, the state of the art of current research and products are discussed. The concept of a smart home; the development of the body-transfer system, body-transfer bed, body-transfer wheelchair, and control architecture; and transfer task scenarios are explained in Section 3. Section 4 shows the user evaluation of the body-transfer smart home system, and Section 5 is a detailed discussion. Finally, Section 6 contains the conclusions.

## 2. State-of-the-Art Assistive Transfer Systems

Numerous studies have emphasized the importance of smart homes as an alternative for AAL of elderly and physically disabled people. Robotic transfer devices have the potential to reduce the efforts and improve the assistance of healthcare workers for a better and more convenient way of transferring patients while simultaneously improving the quality of life (QoL) of elderly and physically disabled people by minimizing their dependence on caretakers [15]. In addition, they can mitigate the complications associated with disabled people and caregivers, i.e., injury, metabolic decline, muscle atrophy, and psychological deterioration. Recently, a plethora of designs for transfer systems have been developed, a few of them being commercially available. However, the frequency of use of such transfer systems is still limited for a few major reasons, such as high cost,

unreliable lifting mechanisms [16–18], frequent manual intervention, and unavailability of a complete transfer system (to and from bed to mobility devices, i.e., wheelchair and toilet chair). According to [19], the concept of maneuvering disabled individuals from beds through a robotic device was initiated by Robot for Interactive Body Assistance (RIBA). The voice-operated robot has an arm with seven degrees of freedom to lift the patient. A wide range of tactile sensors is employed in the system, which strengthens the RIBA to modify the desired trajectory of lifting according to information passed on by the caregiver. However, RIBA has a few functional challenges; for instance, the disabled person may slip from the arms of the RIBA. In addition, the robot cannot lift more than 70 kg of weight. Additionally, the electrical inputs of the robot make it unfit for particular activities such as the bathroom and toilet. Roger and James developed the home lift, position, and rehabilitation (HLPR) chair at the National Institute of Standards and Technology [20,21]. The HLPR robotic system attempts to address the issue of transferring the patient to and from the wheelchair to the toilet. Additionally, it assists in the physical rehabilitation of the patient. Although the system offers wide applications, the bulky and complex structure and lack of maneuverability limit the use of such a system in a home. In addition, transferring the disabled person to the toilet is challenging with the assistance of HLPR. Intelligent Sweet Home (ISH) combines an intelligent bed, a wheelchair, and a robotics hoist. A supporting manipulator is mounted on the bed [22–25]. A communication module and control unit connects and controls all the Intelligent Sweet Home components. The system offers the flexibility of voice and hand gestures as input control. Notably, this intelligent machine system is completely integrated and fully automatic. However, moving the person with a robotic hoist is risky, limiting their use frequency. In addition, a dedicated assistant is required to fasten the belt to the user's body before lifting. A robotic chair/bed system was designed by Ning et al. [26] for the assistance of a bedridden person. It has an integrated U-shaped bed with omnidirectional wheelchairs having the function of reconfigurability. This system has employed a mechanism that allows disabled people to change body posture in bed to avoid certain issues, such as bedsores. However, since it is designed for lower-limb disabled people, a helper is still needed to assist the patient in certain activities, i.e., shifting the disabled person from a wheelchair to a toilet room. Another transfer system, the Patient Transfer System (PTS) [27,28], has also been designed to restore dependencies. It includes a bed and a wheelchair. For transferring the person from the wheelchair to the bed, the caregiver brings the rear wheels of the wheelchair in contact with the docking base, placed underneath the bed. In addition, the caregiver removes the backrest of the wheelchair. Then, the foot deck rises with the help of linear actuators to allow the lower side of the bed (mattress lower end) to come in contact with the back of the person. The wheelchair (seat and footrest) and conveyer belt of the bed rotate in an anti-clockwise direction simultaneously. The foot deck returns to the home position to fully maneuver the person from the wheelchair to the bed. The whole process takes approximately 2–3 min. On one end, the transfer system is reliable and comfortable. On the other hand, continuous supervision by the assistant is needed during the shifting, i.e., the system does not present autonomous relocation from bed to wheelchair and vice versa. A similar kind of concept for transferring the user from the bed to the wheelchair has been proposed [29]. A comparison table is presented in Table 1 to simply show the advantages and drawbacks of each available product.

**Table 1.** Comparison table of the assistive transfer device.

| Name | Year | Transfer Posture | Transfer Mode | Payload (kg) | Assistance in Performing Self-Care Activities | | |
| --- | --- | --- | --- | --- | --- | --- | --- |
| | | | | | Transferring from Bed to Wheelchair and Vice Versa | Toileting | Bathing |
| Robotic smart house to assist people with movement disabilities [23] | 2007 | Lifting | Sitting | 90 | √ | √ | x |
| HLPR chair [20,21] | 2010 | Lifting | Sitting | 93 | √ | √ | x |
| RIBA [19] | 2010 | Lifting | Supine | 80 | √ | √ | x |
| Wheelchair with a lifting function [16] | 2012 | Lifting | Sitting | 150 | √ | √ | x |
| Robotic-assistedtransfer device [17] | 2015 | Lifting | Sitting | 117 | √ | √ | x |
| Robotic chair/bed system [26] | 2017 | Lifting | Sitting | 250 | √ | x | x |
| CarryBot [29] | 2017 | Lifting | Sitting | 57 | √ | √ | x |
| Patient Transfer System (PTS) [27,28] | 2021 | Sliding | Supine | 159 | √ | x | x |
| Novel PersonTransfer Assist System [18] | 2021 | Sliding | Supine | 120 | √ | x | x |
| This system | | Sliding | Supine | 120 | √ | √ | √ |

## 3. The Body-Transfer Smart Home Design Concept

To assist functionally impaired people in the bedroom, bathroom, and home setting, a smart home system that includes a variety of robotic modules has been proposed. Figure 1a shows a general block diagram of the proposed body-transfer smart home system (BTS). All robotic modules are integrated via the central control unit, which is a handheld pendant. This central unit gives a command to each module based on the user's directions. All the robotic modules are designed to perform their inherent functions and work in collaboration. This approach offers many benefits: such modules can be utilized as stand-alone devices, and when connected to another system, their control can be based on user commands. The collaboration of these installed devices results in effective and efficient job execution, expanding the range of daily life activities that the resident can aid.

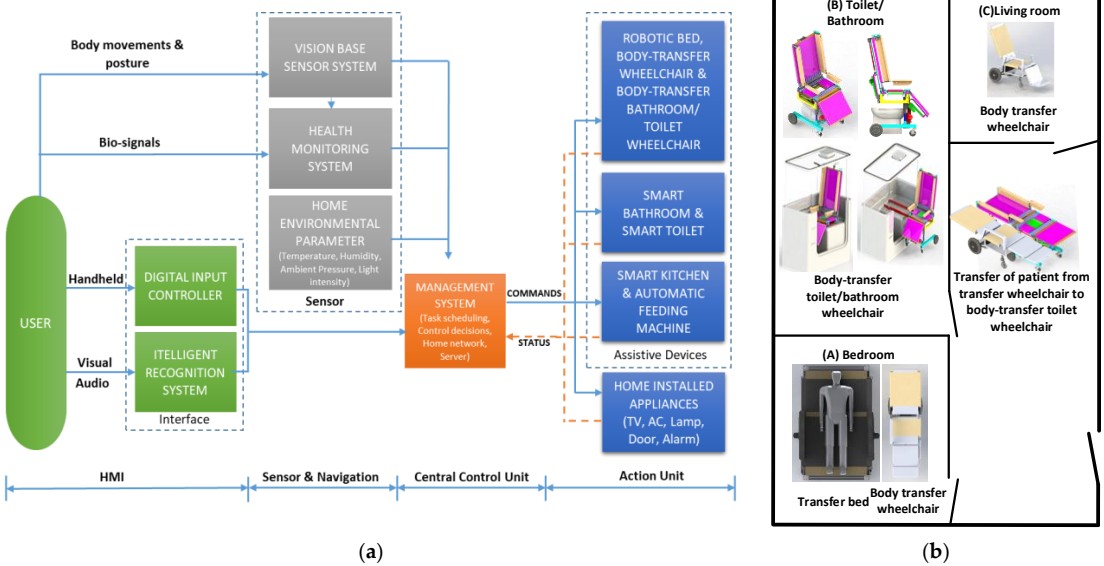

**Figure 1.** (**a**) Smart home end-to-end concept for a disabled person. (**b**) Conceptual illustration of the comprehensive patient transfer system in a body-transfer home.

Figure 1b shows the transfer concept for a user. To transfer a person from a bed to a wheelchair and vice versa, the bed is equipped with conveyor belts. For smooth and secure transfer of the person from the bed to a wheelchair, two conveyor belts function simultaneously. Bruno et al. showed that cost, comfort, construction, handling, and programming complexity are some factors on which several transfer systems have been developed. Thus, based on the decision matrix, a center-driven conveyor belt system is best suited for transferring the person from a bed to a smart wheelchair [30]. A conveyor system is mounted on the wheelchair to receive the person transferred from the bed. The wheelchair's conveyor belts are activated in coordination with the bed's conveyor belts. The simultaneous moving of both conveyor belts transfers the person from the bed to the wheelchair. The same operation is repeated for reverse transfer but in the opposite direction. The wheelchair and the transfer system have a reclining and leg-raising system, which is used for changing the person's posture.

Physically challenged people go through many complications during toileting and bathing, and for this reason, another body-transfer bathroom/toilet wheelchair, as shown in Figure 1b, needs to be developed. This wheelchair is capable of receiving people from the body-transfer bed as well as from the body-transfer wheelchair. The body-transfer bathroom/toilet wheelchair has features similar to those the body-transfer wheelchair, with the additional feature of having a height adjustment mechanism, for accommodating commodes of different sizes. However, as opposed to the powered conveyor belts on a body-transfer wheelchair, it has passive rollers for receiving and delivering the person. These are used to avoid the risk of electrical shock during use in a bathroom. The bathroom has a bathtub mounted with slide rails; these rails assist the caregiver in transferring the person from the body-transfer bathroom/toilet wheelchair to the bathtub.

### 3.1. Proposed Solution

This section describes the proposed comprehensive transfer system in detail. The body-transfer smart home system comprises a body-transfer bed (BTB), a body-transfer wheelchair (BTW), and a body-transfer toilet wheelchair (BTTW).

### 3.1.1. Design of Body-Transfer Bed

The BTB consists of two modules: the transfer module and the safety module. Figure 2a shows the CAD model of the BTB. Note the two endless conveyor belts mounted on the bed. Two centered AC servomotors mounted opposite to each other drive these conveyor belts.

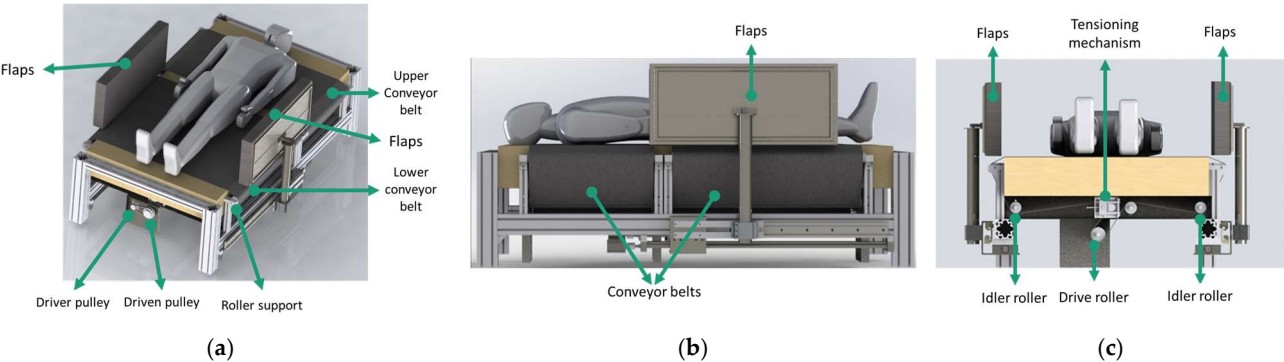

**Figure 2.** Body-transfer bed: (**a**) isometric view; (**b**) right view; (**c**) front view.

The specification of the BTB frame is 2075 × 1150 × 580 mm, whereas the size of the BTB is 1880 × 1060 × 200 mm. Two conveyor belts have been mounted on the BTB for transferring the person from the bed to both wheelchairs. The conveyor belts have been designated as upper and lower, their sizes being 3600 × 590 × 3 mm and 3600 × 900 × 3 mm, respectively. To maintain proper hygiene, they are covered by a cloth that acts as a bedsheet via Velcro. Two friction rollers, their lengths 606 mm and 910 mm, respectively, drive both the upper and lower conveyor belts. The diameter of both rollers is 76.5 mm. Figure 2b

depicts the drive mechanism of conveyor belts: the center-driven conveyor belt passes through a series of idler rollers; rollers A and B are crown rollers, which keep the belt trajectory in a straight line. Additional 500 mm idler rollers have been mounted on both sides of the BTB to reduce the friction between the belt and the bed along the sides. During the transfer, since the patient's body has to pass over the rollers, the same has been covered with a 75 mm thick foam so that the patient will be comfortable while passing over the rollers. A spring-loaded tensioning mechanism has been placed in the drive mechanism to keep both conveyor belts under tension.

Roller supports have been bolted with a horizontal sliding frame to keep the drive rollers vertical, as shown in Figure 2c. Since the direction of motion of the conveyor belts is bi-directional, there is a danger posed to the physically disabled person—caused by the accidental movement of the belts—which might result in serious injury to the patient. Hence, the inclusion of a safety module is inevitable. This module comprises a motorized guide rail, a double rail, and a motorized protection flap, as shown in Figure 2b,c. The function of the motorized guide rail is to move the flap out of the way during the transfer process. Once the transfer is completed, it returns to its original position to secure the person lying on the BTB. It has been bolted on the lower side of the BTB frame. For reinforcing and countering the unbalanced forces acting on the flap, the double rail has been attached to the carriage of the guide rail using an L-shaped bracket. The guide rail has a travel stroke of 1000 mm and is powered by a NEMA23XL stepper motor. Limit switches have been placed on both sides of the guide rail to stop the motor. Similarly, the double rail has a length of 1000 mm. A motorized flap frame has been mounted on the L-shaped bracket using a pipe, having a flange for holding the motor, which rotates the flap frame. The flap's motor is a DC-geared motor whose rotation is controlled by two limit switches bolted firmly onto the flange.

### 3.1.2. Design of Body-Transfer Wheelchair

This study aims to reduce the patient's dependence on caregivers and allow the patient to be transferred from the BTB to other mobility devices independently. For receiving and transferring patients, the wheelchair should have the capability of fully reclining and leg raising. In addition, it was found that instead of building a new wheelchair from scratch, it is better to refit a commercially available wheelchair with a transfer device similar to a BTB. The model used in the project is the Merits J610H fully reclining rehab wheelchair [16]. Most commercially available powered wheelchairs can only recline to a limited degree. The same is the case with the leg-raising feature. Moreover, a powered wheelchair with full reclining capability is very expensive, which restricts its usage. The seat dimensions of the wheelchair are as follows: height of the backrest, 800 mm; depth and width of the seat, 420 and 550 mm, respectively. The weight capacity of the wheelchair is 114 kg. Three mechanisms are installed on the wheelchair in this system: a reclining and leg-raising mechanism, a transfer mechanism, and an autonomous docking system.

### 3.1.3. Reclining and Leg-Raising Force Evaluation

Reclining and leg-raising features are achieved by placing an actuator beneath the seat shown in Figure 3a. The moment equation is applied to the connections between the backrest and the seat to evaluate the force required for pushing the actuator. Analysis of forces acting on the backrest and leg rest evaluated using a free body diagram is shown in Figure 3b,c.

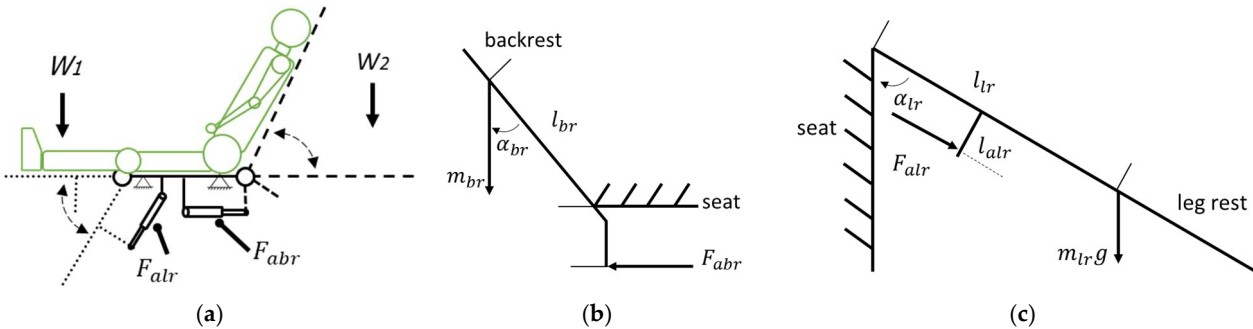

**Figure 3.** (**a**) Reclining and leg-raising concept. (**b**) Free body diagram of the backrest. (**c**) Free body diagram of the leg rest.

The moment of the backrest is evaluated using a free body diagram shown in Figure 3b, and the balance of the moment can be written as in the following equation:

$$M_{br} = 0 = -l_{br} m_{br} g \sin \alpha_{br} + l_{abr} F_{abr} \tag{1}$$

where $M_{br}$, $m_{br}$, $F_{abr}$, $l_{br}$, $l_{abr}$, $g$, and $\alpha_{br}$ are the total moment, the total weight of the backrest and weight acting upon it, actuator force, the distance between the connection point and the point the weight is acting upon (440 mm), the distance between the connection point and $F_{abr}$ (100 mm), gravitational acceleration, and the angle between the backrest and the total weight, respectively. The force of the linear actuator, $F_{abr}$, can be written as in the following equation:

$$F_{abr} = l_{br} m_{br} g \sin \alpha \frac{1}{l_{abr}} \tag{2}$$

The center of gravity for the body weight acting upon the backrest and leg rest changes according to the position of the body parts. However, since the backrest and leg rest are covering a wider area, the center of gravity for the backrest and leg rest is farther from the center of gravity for the body weight. Thus, choosing the same position for the centers of gravity of body weight with backrest and leg rest weight results in a safety margin for all the positions where all the body parts are in the wheelchair.

For a 100 kg person, the upper body's weight, which rests on the backrest, can be calculated as in Equations (3) and (4):

$$m_{upperbody} = m_{head} + m_{throax} + m_{abdomen} + m_{total\,arm} \tag{3}$$

and

$$m_{br} = m_{upperbody} + m_{backrest} \qquad m_{br} = 44.77 \text{ kg} + 6 \text{ kg} = 50.77 \text{ kg} \tag{4}$$

Similarly, the leg-rest free body diagram is shown in Figure 3c. The moment equation of this system around the connection between the seat and leg rest can be written as in the following equation:

$$M_{lr} = 0 = l_{lr} m_{lr} g \sin \alpha_{lr} - l_{alr} F_{alr} \tag{5}$$

where $M_{lr}$, $m_{lr}$, $F_{alr}$, $l_{lr}$, $l_{alr}$, $g$, and $\alpha_{lr}$ stand for the total moment, the total weight of the leg rest and weight acting upon it, actuator force, the distance between the connection point and the point the weight is acting upon (280 mm), the distance between the connection point and $F_{act}$ (30 mm), gravitational acceleration, and angle, respectively. The moment equation of this system is around the connection between the seat and between the seat and leg rest backrest.

So, the force of the linear actuator, $F_{alr}$, can be evaluated using the following equation:

$$F_{alr} = l_{lr} m_{lr} g \sin \alpha_{lr} \frac{1}{l_{alr}} \tag{6}$$

For a 100 kg person, the weight of the legs and the feet which rests on the leg rest can be calculated using the following equation:

$$m_{lr} = m_{leg\&foot} + m_{legrest} \tag{7}$$

$$m_{lr} = 6.43 \text{ kg} + 2 \text{ kg} = 8.43 \text{ kg}$$

The result of Equations (2) and (6) is represented in the graph of the actuator force change required to move the backrest and leg rest according to the angle as shown in Figure 4. Thus, the maximum force for the backrest $F_{alr}$ and leg rest $F_{br}$ occurs at $\alpha = 90°$, calculated as 2192 N and 772 N, respectively. Based on this information and considering the factor of safety 2, the force required by the actuator to move the backrest is 4384 N. However, the force applied by the actuator on the leg rest should be 1544 N.

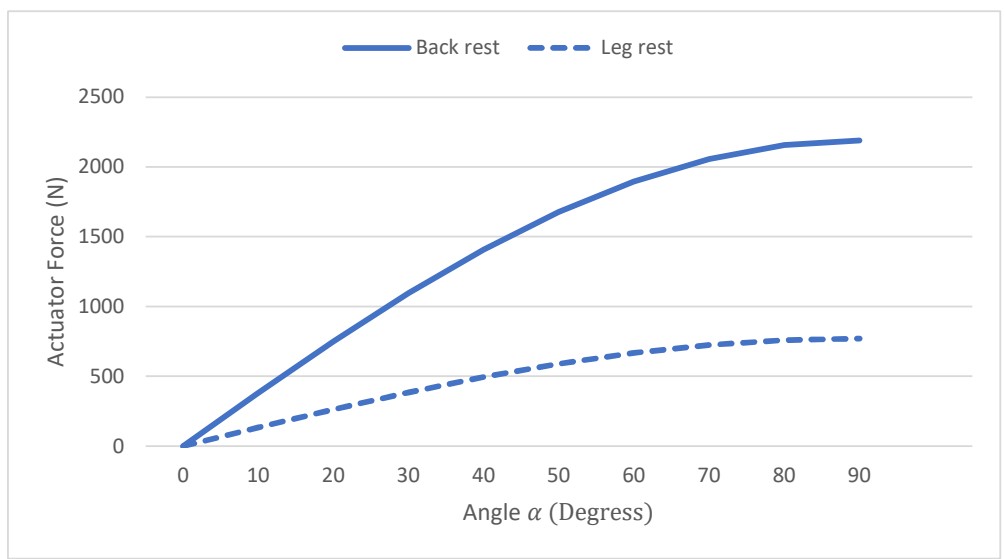

**Figure 4.** Backrest and leg-rest actuator force variation with the change in angle.

### 3.1.4. Transfer Mechanism

The transfer mechanism plays a critical role when the patient is transferred laterally from the bed to the wheelchair and vice versa. To achieve this, a mechanism consisting of two conveyor belts has been designed, as shown in Figure 5a,b.

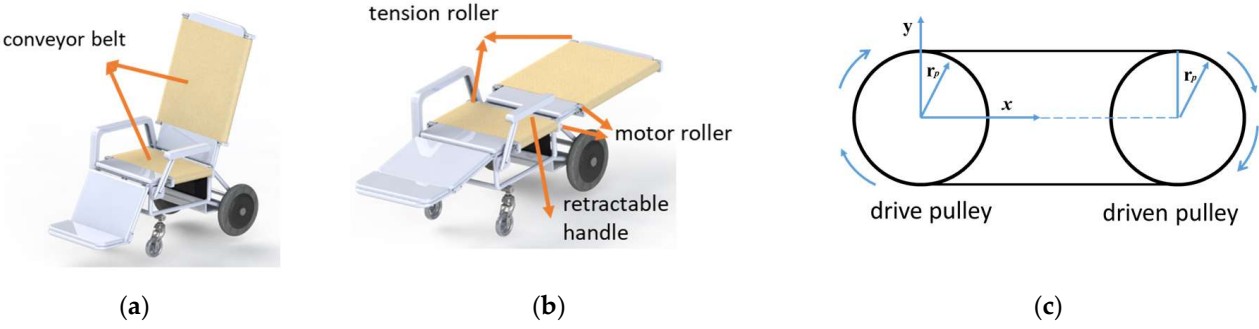

(**a**)   (**b**)   (**c**)

**Figure 5.** CAD design of BTW (**a**) BTW in normal position (**b**) BTW in recline position (**c**) Free body diagram of belt drive system.

The drive mechanism of the conveyor belts should be in such a way that the conveyor belts pass over the mattress of the wheelchair. Moreover, for lateral transfer setup, there should be minimum space between the bed and wheelchair rollers. The reduction in space between the rollers was a key issue. Considering this problem, it was decided to use drum motor rollers. Furthermore, these motor rollers not only operate more quietly, but they are

more energy efficient and more compact compared to gear motor-driven rollers. They are also easier to install and maintain.

In a belt drive system, continuous torque is calculated by taking the mean root of all the torque requirements throughout the application, the torque required for acceleration, constant velocity, and deceleration. In most applications, the maximum (intermittent) torque occurs during acceleration [31]. For a belt drive system, as seen in Figure 5c, the motor torque required during constant velocity is simply the total axial force ($F_a$) on the belt multiplied by the radius of the drive pulley as shown in Equation (8), where $T_C$, $F_a$, $r_1$, and η are torque required during constant velocity (Nm), total axial force (N), the radius of the drive pulley (mm), and efficiency of the belt drive system, respectively.

$$T_c = \frac{F_a r_1}{1000\eta} \tag{8}$$

where

$$F_a = m_{ml}g\mu \tag{9}$$

In Equation (9), $m_{ml}, g$, and $\mu$ are the mass of the moved load (external load plus belt) (kg), acceleration due to gravity (m/s$^2$), and coefficient of friction of the guide, respectively. The total torque required $(T_a)$ includes the torque required at constant speed plus the torque required to accelerate the load.

$$T_a = T_c + T_{acc} \tag{10}$$

$$T_{acc} = J_t \alpha \tag{11}$$

where $T_c$, $T_{acc}$, $J_t$, and $\alpha$ are torque at constant speed (Nm), torque required due to acceleration (Nm), total inertia of the system (kgm$^2$), and angular acceleration (rad/s$^2$), respectively. The motor drive torque required for deceleration is equal to the torque at constant velocity minus the torque due to acceleration, written as follows:

$$T_d = T_c - T_{acc} \tag{12}$$

where in $T_d$ is the torque required during deceleration (Nm). Thus, the continuous torque required by the motor can be calculated as follows:

$$T_{RMS} = \frac{\sqrt{T_a^2 t_a + T_c^2 t_c + T_d^2 t_d}}{t_{total}} \tag{13}$$

where $T_{RMS}$, $t_a, t_c, t_d$, and $t_{total}$ are root mean square (continuous) torque (Nm), time for acceleration (s), time for constant velocity (s), time for deceleration (s), and total time for the movement (including any idle time between movements) (s), respectively.

A drum motor roller is a roller that has an internal motor [32]. This motor has a compact design, saving much space compared to conventional conveyor designs. The roller motors' specifications were evaluated using a safety factor of 2, resulting in the backrest conveyor's required torque of 11.63 Nm and a minimum speed of 13 rpm on the 50 mm diameter of the roller. The seat conveyor required a torque of 8.4 Nm and a minimum speed of 13 rpm on the 50 mm diameter of the roller. Based on the roller motor calculation, a roller motor has been selected to drive the conveyor belts to drive the backrest and seat, as shown in Table 2.

**Table 2.** Specifications of the roller motor.

| Rated Power (Watts) | Gear Box Series (Stage) | Rated Speed (RPM) | Allowable Torque (Nm) | Driving Force (N) |
|---|---|---|---|---|
| 40 | 3 | 12.90 | 15 | 1200 |

Figure 6a shows, the installed conveyor mechanism on BTW matching with the conveyor mechanism of BTB. The flat belt used in this system is made by MITSUBOSHI (NS41UF0/2W, MITSUBOSHI, Japan) engineering. The belt is made of pure natural rubber that is pollution-free and safe for human use [18]. A hand controller on the wheelchair controls the conveyor belts on the wheelchair and bed. The BTW communicates with the BTB using an NRF24L01 wireless transceiver.

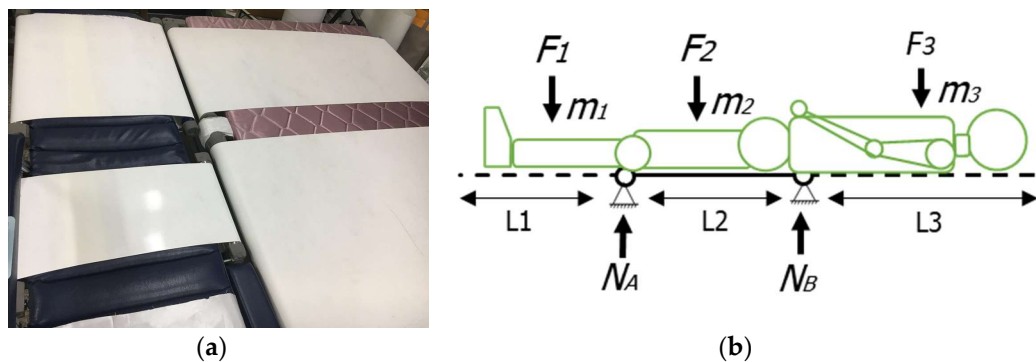

|  (**a**)  |  (**b**)  |
|---|---|

**Figure 6.** (**a**) BTW docked next to BTB (**b**) Force distribution for tipping analysis.

According to Figure 6b, the tipping/stability problem can be avoided if the force at the point $N_A$ and $N_B$ is positive. The moment acting on the $N_A$ and $N_B$ can be defined as follows:

$$\sum M_A = 0 \tag{14}$$

$$-F_1 \frac{L_1}{2} + F_2 \frac{L_2}{2} + F_3 \left(L_2 + \frac{L_3}{2}\right) - N_B L_2 = 0$$

$$\sum M_B = 0 \tag{15}$$

$$-F_1 \left(L_2 + \frac{L_1}{2}\right) - F_2 \frac{L_2}{2} + F_3 \frac{L_3}{2} + N_A L_2 = 0 \tag{16}$$

where $m_1$ is the mass of leg and leg rest, $m_2$ is the mass of the thigh and middle part of the wheelchair, and $m_3$ is the mass of the upper limb and backrest. According to the specifications of the wheelchair, $L_1$, $L_2$, and $L_3$ are 560, 400, and 880 mm, respectively. The masses applied to the system $m_1$, $m_2$, and $m_3$ are 24.2, 42.3, and 50.77 kg, respectively. From Equations (14) and (15), the forces acting on points $N_A$ and $N_B$ are 67.6 N and 1083 N, respectively. The positive results indicate that the wheelchair will not tip to the back or front sides.

### 3.1.5. Hand rest Mechanism

The BTW has a movable hand rest for securing the patient. However, to enable the lateral movement of the patient, it is necessary to remove the hand rest so as not to obstruct the motion while also providing support during the transfer process. To accommodate the obstruction, a rotating hand rest is installed on one side of the BTW, designed to move out of the way before the transfer starts. The backrest and the seat has become wider, as compared to the leg rest, because of the transfer mechanism, as shown in Figure 6a. So there should be support under all body parts during transfer. Two rotating bars were designed to move the hand rest so that the hand rest always stays parallel to the ground which is depicted in Figure 7a,b. A 24 VDC (volts of direct current) worm gear motor with a rated torque of 170 kg-cm having a speed of 30 rpm was selected for controlling the hand rest movement because of its self-locking capabilities. Limit switches were placed at the desired locations to control the movement of the hand rest.

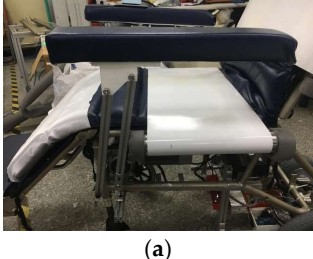
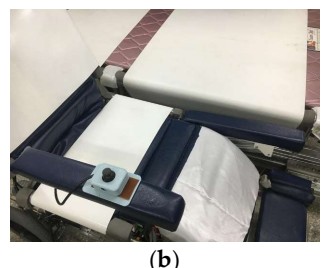

(**a**)  (**b**)

**Figure 7.** (**a**) Hand rest in the normal position. (**b**) Hand rest in the lowest position.

3.1.6. Wheelchair Drive Mechanism

For turning the manual wheelchair into a powered one, two wheels with a hub motor were added to the wheelchair. The wheels are assembled so that they also provide support against tipping over. The wheels chosen are equipped with 24 VDC brushless hub motors with a rated power of 200–250 watts and a speed of 8 km/h. The motors are controlled by a joystick.

A handheld controller connected to the wheelchair was designed and developed to control the system on the wheelchair and bed. The buttons and their designated functions are listed in Figure 8a. As mentioned beforehand, the bed has two flaps on the sides. These flaps provide safety measures for the user during transfer and when the person is in bed. While one of the flaps is stationary, as shown in Figure 8a, the other can be controlled from the wheelchair to move to an open position to clear the way of transfer. After pushing button 10 once on the controller, the flap will start moving towards the end of the bed until it reaches the extremity where it will hit a limit switch. This causes the flap to turn around to clear the way, as shown in Figure 8c.

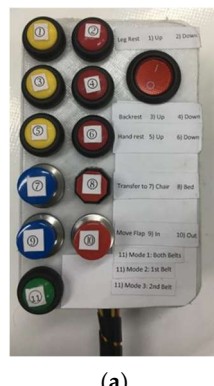
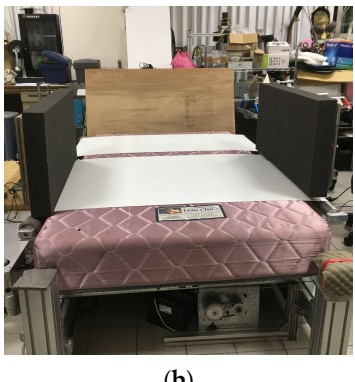
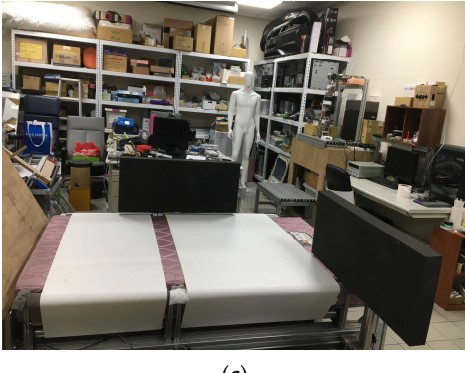

(**a**)  (**b**)  (**c**)

**Figure 8.** (**a**) Handheld controller. (**b**) Flaps in the closed position. (**c**) Flaps in the open position.

After the flaps are open, the wheelchair can be placed next to the bed, as shown in Figure 9a. Afterward, by pushing button 6, the handrest can be lowered, as depicted in Figure 9b. Later, for the transfer to begin, the wheelchair should be fully reclined from the chair position. Pushing button 1 on the controller raises the leg rest while pushing button 4 lowers the backrest. Pushing the buttons until the backrest and the leg rest reach their extremities results in the wheelchair reaching a fully reclined position, ready to transfer the user. The fully reclined wheelchair can be seen in Figure 9c. When the wheelchair is in position, pushing button 8 starts the conveyor belts on the wheelchair and bed, and the user can be transferred from the wheelchair to the bed. After the user is transferred to bed, the wheelchair can be removed, and the flap can be moved inside by pushing button 9. Similarly, the user can be transferred back to the wheelchair from the bed in reverse order.

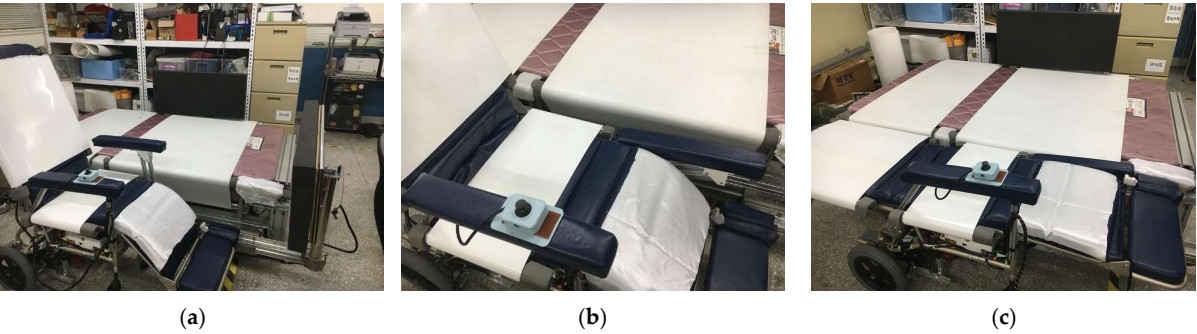

(**a**)    (**b**)    (**c**)

**Figure 9.** (**a**) Wheelchair in position next to the bed. (**b**) Handrest in the lowered position. (**c**) Fully reclined wheelchair next to the BTB.

### 3.1.7. Body-Transfer Toilet Wheelchair (BTTW)

The BTTW is designed to assist aged people in performing bathing and toileting tasks safely and conveniently. The BTTW has a powered leg rest and backrest mechanism that can be rotated from 180° to 200° and from 0° to 70°, respectively. This feature allows the patient to change the posture in the wheelchair from regular sitting to a fully supine position, as shown in Figure 10a,b. Since the BTTW is to be used in wet areas, the seat, backrest, and leg rest of the BTTW are made up of a series of passive rollers. This considerably reduces the risk of electrocution in the toilet/bathroom. The seat has a sliding commode slot which helps the user to perform toileting while sitting on the seat. The commode slot can be easily operated by pulling/pushing the commode slot handle, as shown in Figure 10c,d. Another purpose of these rollers is to rapidly clean and dry the seat, backrest, and leg rest after being utilized in the toilet/bathroom. The BTTW is not designed for prolonged use. Furthermore, the seat can be detached from the mainframe so that bathing can be performed [33].

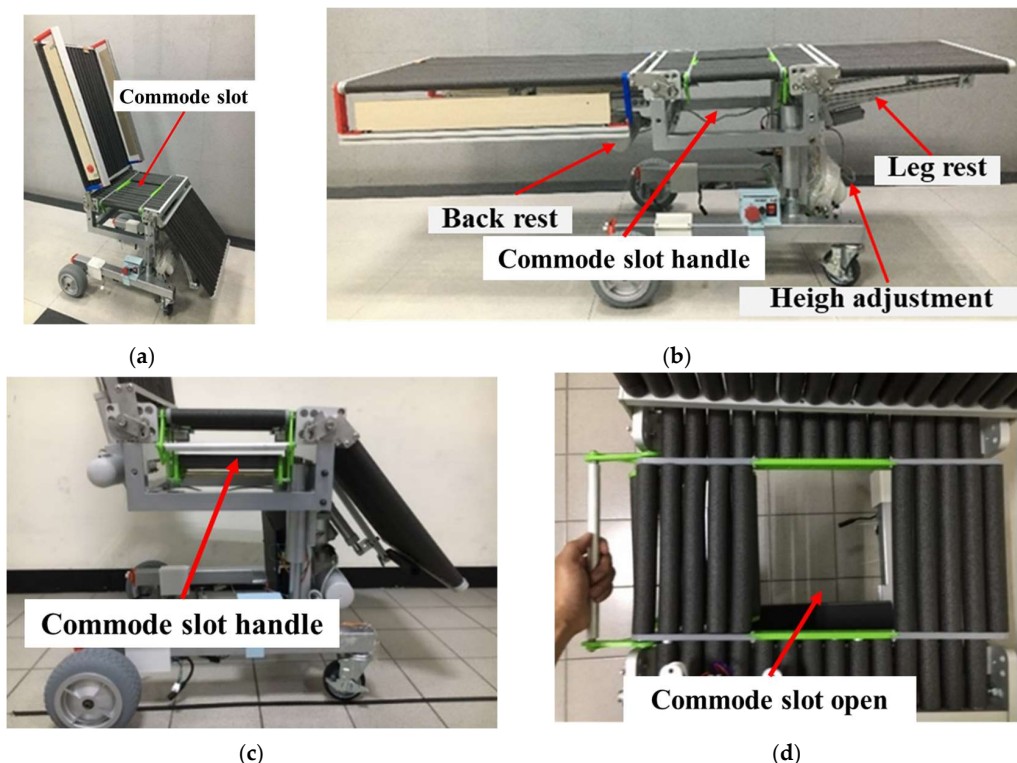

(**a**)    (**b**)

(**c**)    (**d**)

**Figure 10.** Body-transfer toilet wheelchair: (**a**) isometric view; (**b**) supine position; (**c**) commode slot open position; (**d**) opening commode slot [33].

### 3.1.8. Operational and Control System

Figure 11 shows an operational diagram to manage all activities in the wheelchair. An integrated control algorithm has been developed to ensure the user's safety while transferring from the bed to both wheelchairs. It starts with the input of the user. Initially, the bed safety module is activated and checks the position of the guide rail with the help of position sensors mounted on both ends of the guide rail. Once the safety flaps have been opened, the docking module ensures that the wheelchair has been docked. If the wheelchair is not docked, then there will be a bed safety alarm that alerts the caregiver. After the docking of the wheelchair is confirmed, the wheelchair safety module starts to operate.

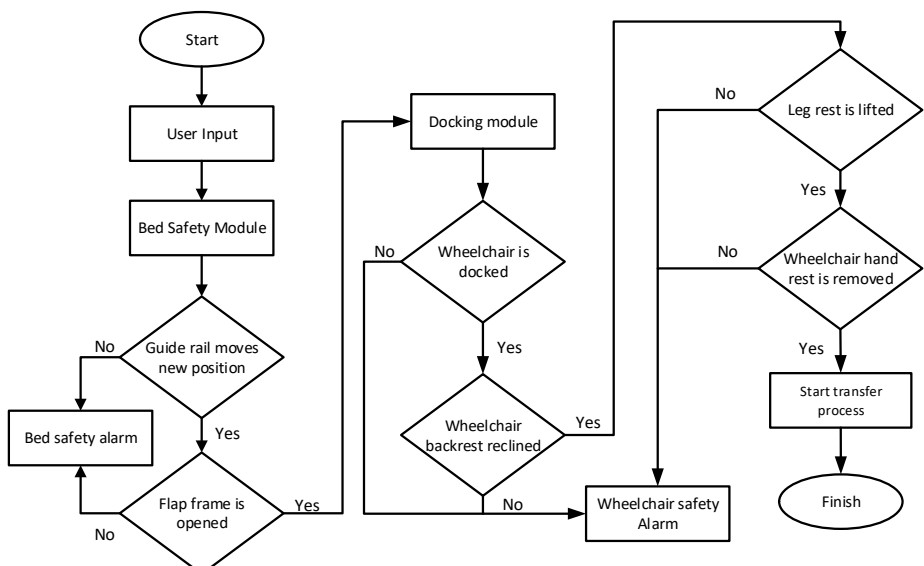

**Figure 11.** Operational diagram.

### 3.1.9. Control Architecture of the Transfer System

The human–machine interface employed in a body-transfer smart home design will significantly impact whether the user will embrace it. This study concentrated on designing a "human-friendly interface" which conveniently enables human–machine interaction. Various technical solutions have been explored to provide a straightforward method of transferring information between the user and the devices installed in the home. Voice- and gesture-based interfaces are frequently employed [34]. There are no trustworthy techniques for voice recognition in a noisy environment. Therefore, a handheld pendant with a human-friendly microcontroller has been developed.

Figure 12 shows the control architecture of the transfer system. The direct line indicates the wire communication between devices, and the dashed line indicates the communication between devices using Bluetooth. In this case, it only focuses on three assisting devices, namely the BTTW, BTW, and BTB. A 24 VDC battery powers both wheelchairs. The Arduino Mega 2560 is the main controller managing logic and communication between the various sensors, actuators, command input, and display indicators in both wheelchairs. The different features of the BTTW and BTW are the bathroom mode and conveyor actuator, respectively. A handheld pendant is used to control both wheelchairs and execute the program while in the bathroom mode or during docking to the BTB. For safety reasons, a battery temperature sensor and alarm are used on each wheelchair. Furthermore, the bed sensing and actuators, such as conveyor belts and motor flaps, are controlled using Raspberry Pi 4. The bed has two different power lines; the conveyor uses a 110 volts alternating current (VAC) power driver, and the driver flap uses 24 VDC.

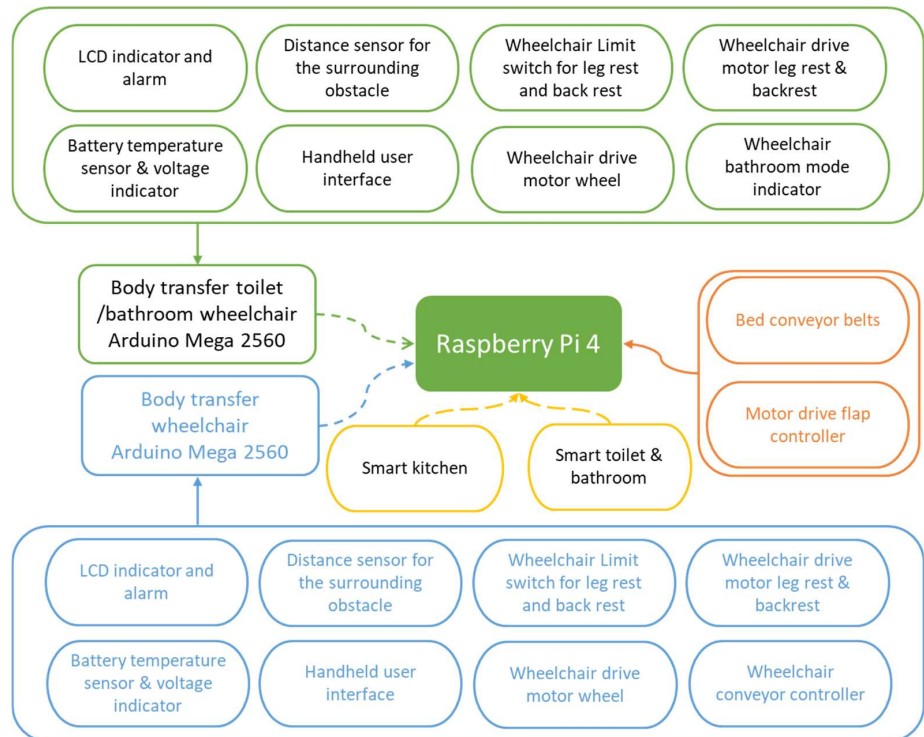

**Figure 12.** Control architecture of the BTS.

*3.2. Task Scenarios*

The three crucial operations of the proposed design strategy for the BTS were performed to complete the prototype verification. Firstly, a person was transferred from a BTB to a BTW (task scenario 1). Secondly, a person was transferred from the BTB to the BTTW (task scenario 2). Finally, the transfer of a person from the BTW to the BTTW (task scenario 3) was evaluated. Furthermore, all the activities on the system were performed under the supervision of a caregiver. The various task scenarios of the BTS are shown schematically in Figure 13.

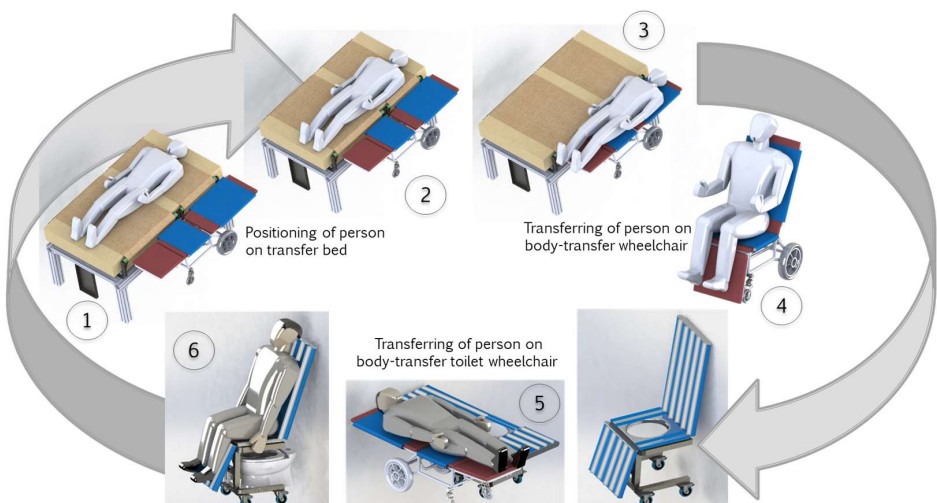

**Figure 13.** Transfer task scenarios (1) Positioning of person on the transfer bed (2–4) Transferring of person on the BTW (5) Transferring of person on BTTW (6) A person perform toileting while sitting on BTTW.

### 3.2.1. Task Scenario 1: Transfer from Body-Transfer Bed to Body-Transfer Wheelchair and Vice Versa

The most significant task for people with limited mobility is the transfer to and from a BTB. Figure 14a–e depict the user's progression from the BTB to the BTW. Figure 14a shows that the BTW is brought to the side of the docking base for transfer. Afterward, the safety guide rail is brought to the end of the BTB to ensure that the person can be transferred safely and smoothly. When the guide rail reaches the extreme point, the flap motor opens the flap, as shown in Figure 14b. In Figure 14c, it can be seen that the BTW's backrest reclined to its extreme points, and the wheelchair's leg rests are simultaneously brought up so that it becomes a stretcher. When both systems are ready for transfer, the caregiver/person being transferred gives a signal to initiate a transfer via pendant.

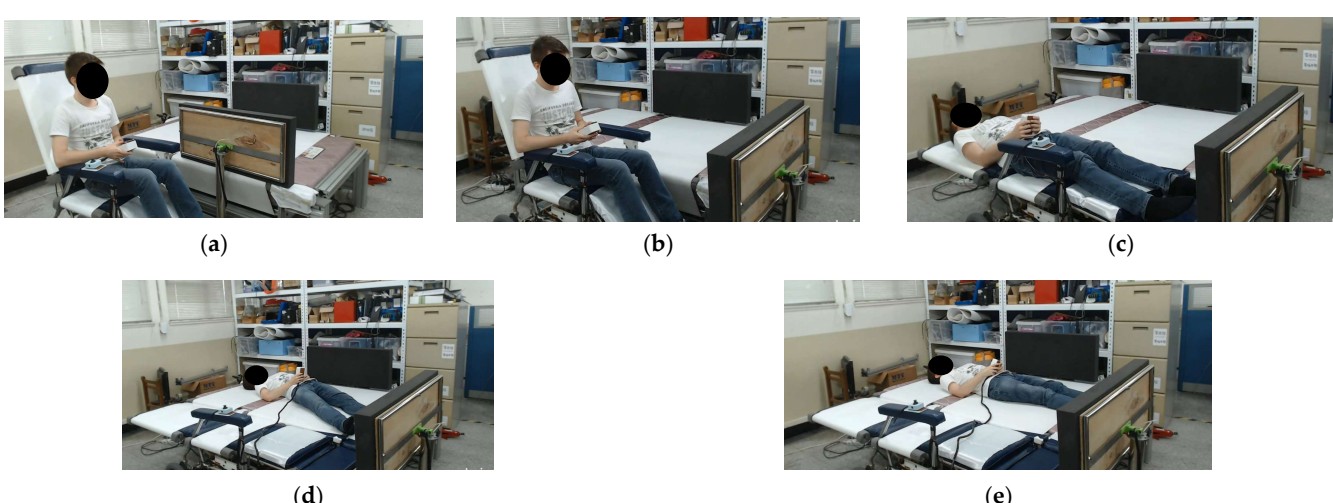

**Figure 14.** Transfer from BTB to BTW and vice versa (**a**) Person sitting on the BTW positioned next to BTB (**b**) Person remove the safety flap on BTB (**c**) BTW in reclined state (**d**) Person transferring from BTW to BTB (**e**) Adjustment of posture on the BTB.

Figure 14d shows the conveyor belts of both systems running simultaneously at the same speed. Figure 14e demonstrates that once the person is transferred from the BTB to the toilet wheelchair or vice versa, the person can stop the transfer process, and if the body posture is disturbed, then it can be aligned with the help of two independently controlled belts. While moving from the wheelchair to the BTB, the steps mentioned above are repeated. After transfer, the wheelchair returns to the charging station. The sequence to transfer the person from the BTB to the wheelchair and vice versa is shown in Figure 14.

### 3.2.2. Task Scenario 2: Transfer from Body-Transfer Wheelchair to Body-Transfer Toilet Wheelchair and Vice Versa

The transfer situation from a BTW to a BTTW is shown in Figure 15a–d. As illustrated in Figure 15a, the caregiver sets the BTTW next to the BTW. For ease of transfer, both wheelchairs are reclined, and the subject is transferred in a supine state. After approaching the wheelchair, as depicted in Figure 15b,c, the caregiver gently pulls the person's body onto the wheelchair to assist with the transfer. Since the toilet chair has a passive transfer mechanism, the caregiver has to assist with the transition. As seen in Figure 15d, the subject is finally transported to the BTTW.

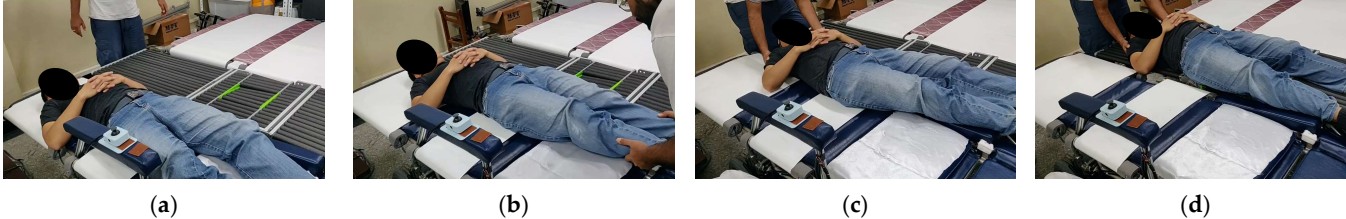

**Figure 15.** Transfer from BTW to BTTW and vice versa (**a**) BTW is positioned next to BTTW (**b**) Caregiver move lower portion of the person's body to BTTW (**c**) Caregiver move upper portion of the person's body to BTTW (**d**) Caregiver repositions the person on the BTTW.

3.2.3. Task Scenario 3: Transfer from Body-Transfer Bed to Body-Transfer Toilet Wheelchair and Vice Versa

The transfer scenario of a person from a BTB to a BTTW is the same as task scenario 1, except for the transfer process. Since there is a passive transfer system on the toilet chair, the caregiver must be involved in the transfer process, as shown in Figure 16a–f. When the person is partially transferred to the BTTW, the caregiver needs to provide a gentle push to transfer the person, as shown in Figure 16d,e. Similarly, in reverse transfer, the caregiver needs to push the person to transfer the person's weight on an active conveyor system [33].

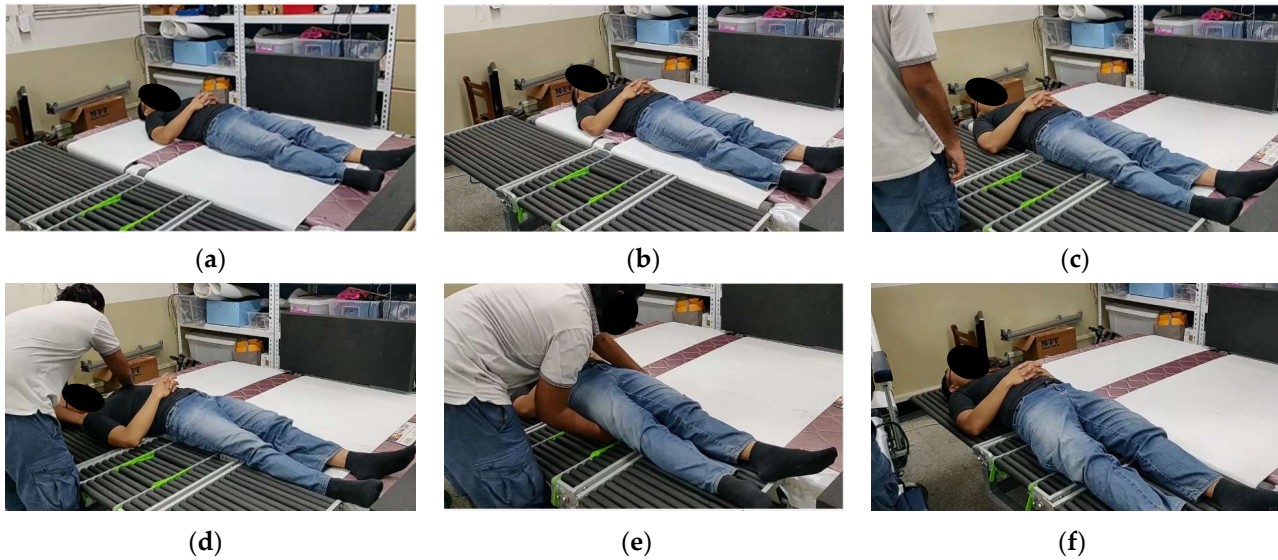

**Figure 16.** Transfer from BTB to BTTW and vice versa [33] (**a,b**) Positioning of the person on the BTB (**c**) Conveyor belt start transferring the person to BTTW (**d,e**) Caregiver moving the upper and lower portion of person's body to BTTW (**f**) Repositioning a person's body on the BTTW.

## 4. User Evaluation of the Body-Transfer System

Two focus groups were chosen to evaluate the effectiveness and acceptance of the BTS in terms of ease of transfer, safety, comfort, stability, and aesthetics, as well as in reducing the load on caregivers and persons with functional impairments when transferring. Wheelchair users and trained caregivers were participants in these focus groups. Of the 20 people who had been using wheelchairs for at least five years, 12 of them were males and 8 of them were females. Seven individuals were using an electric wheelchair, while 13 participants were using a manual wheelchair. Seven professional caregivers with at least a year of transfer experience participated in a different focus group.

After the participants had given their informed consent, a brief survey was distributed to them. Basic demographic information, including age, gender, and ethnicity, and details on the kind of handicap, the number of years of transfer experience, and the assistive devices utilized were included in this survey. Both focus groups received a thorough

presentation on the functions and features of the BTW. They completed another survey after the presentation. This study was designed to learn the people's overall opinions of the BTW.

The survey statements for both functionally impaired people and caregivers included the following: (1) I would like to use the BTS for getting transferred. (2) It is convenient to learn and operate the BTS. (3) The system enables users with lower extremity disabilities to transfer independently. (4) Using a BTS as my aid would make me unsafe. (5) Using a BTS as my aid would make me feel ashamed. (6) It would be easier to get another person to assist in transferring and using the bathroom and toilet rather than using the BTS. (7) such assistive technology capable of assisting in transfers must be built in the future. The result is represented in Figure 17, which is a comparison of responses between the functionally impaired people (U) and caregivers (C).

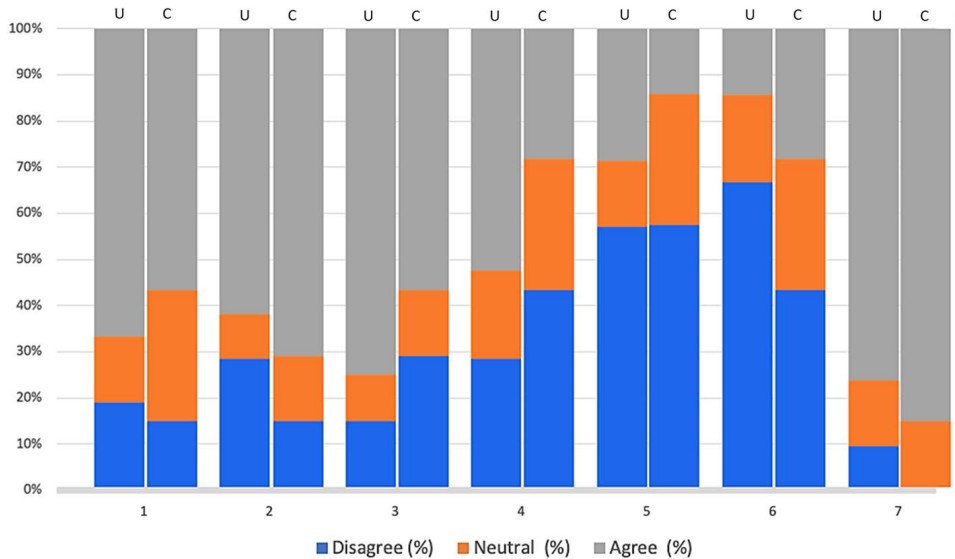

**Figure 17.** Survey of functionally impaired people and caregivers after the presentation regarding body wheelchairs.

Using a 7-point Likert scale, the results were examined. When asked to rate the relevance of a question, responses of 1 to 2 were considered insignificant, 3 to 5 were neutral, and 6 to 7 were considered substantial. The same method was applied to questions that asked respondents to rate their level of agreement with a statement: 1 to 2 indicated disagreement, 3 to 5 indicated neutrality, and 6 to 7 indicated agreement.

Figure 17 reveals the post-presentation poll's findings on the general opinions regarding the BTS. Sixty-seven percent of individuals with functional limitations said they would be open to using a BTS. However, only 14% of those surveyed said they were neutral, and only 14% said they would not use the suggested system. Similarly, 61% of respondents said it is helpful and extremely handy in their everyday lives. All participants believed that the creation of this wheelchair was significant, and 76% said that both the public and commercial sectors should invest resources in furthering the development of the BTS. Participants evaluated the equipment's ease of transfer, safety, and stability as its most critical features, while aesthetics were considered its least relevant attribute.

The caregiver focus group was made up of five men and two women. The participants were 35 years old on average. While the other caregivers supported their clients using manual wheelchairs, one aided a person who used both a manual and a motorized wheelchair. Three of the caregivers utilized motorized lifts, while two of them used hydraulic lifts. Forty percent of caregivers had fewer than five years of transfer-related experience. Two caregivers claimed to have been injured during transfers. For transfer from a BTB to a wheelchair to assist the person in toileting and showering, only one caregiver would prefer

not to use the BTS; the others would much rather use the prototype. Most of the caregivers had no opinion about the visual element. Every participant agreed that creating this type of assistance system was necessary, as shown inFigure 17.

## 5. Discussion

End users and caregivers noticed a reduction in their perceived physical workload while using the BTS. The BTS sliding transfer solution lessens the physical effort required to prepare for and carry out transfers. The end user can be repositioned on the mattress without human/manual lifting with the help of two independent conveyor belts. Repositioning a person in bed frequently or after a wheelchair-to-bed transfer is extremely physically challenging and often results in musculoskeletal pain and injuries [35]. Additionally, manual repositioning procedures resulting in pain and injuries are performed at home because of cost and space constraints [15]. Notably, 25% of accidents in healthcare settings occur during bed transfer [36]. The BTS lessens physical strain during both transfers, mitigating the risk of accidents and repositioning by automating the repositioning process. Studies have shown that mobility has been highly correlated with active assistive living, and its outcomes have been linked with health-related issues [37]. Care receivers have been proven to experience physical and psychological stress during transfers [38]. Therefore, this technique could lower the frequency of musculoskeletal injuries and the physical strain caregivers feel as there are both formal and informal caregiving individuals [39,40].

The majority of the elderly will end up in wheeled mobility devices [41]. For care providers, wheelchair-bound patients are potentially the most dangerous due to wheelchair maneuvering and the process of transfer to the toilet or bathroom, contributing to more than 30% of accidents filed [42]. The lifting method is riskier because the bathrooms can cause someone to slip and fall during their use [43]. The transfer process from the conventional wheelchair to the bathtub and toilet has a high risk of failure due to the lifting method. The caregiver also has some physical pain in their own body while lifting the patient. The proposed method provides a solution by delivering a smooth transfer for the patient without requiring effort on the part of the caregiver or the patient. All the processes are automatically performed by the wheelchair maneuver and structural configuration. During toileting, the wheelchair needs to move back to any size of sitting toilet and open the commode slot easily. Bathing can be performed by sliding the seat and separately detaching it from the main frame while the patient still sits on it. Furthermore, the sliding design and method also have fewer electronics to eliminate the risk of electronic malfunction while the bathroom tasks are performed in a wet area. Additionally, when a person who needs assistance with transfers loses that support (such as an aged spouse), institutional care is frequently recommended (e.g., nursing home). By offering an assistive technology solution that is simple to use in a residential setting, people and the people who care for them may be able to stay in their homes longer and continue to play an active role in their communities.

## 6. Conclusions

Recently, assistive technology has received remarkable attention from researchers due to the exponential growth of elderly, disabled, and bedridden patients. Though the previously designed approaches resolved the basic patient healthcare issues, they are still deficient in transferring patients efficiently. According to the demand of global healthcare systems, a human-centered design and an integrated assistive mechanism design are required. This study proposed a novel sliding transfer approach for transferring the user from bed to wheelchair and vice versa. Furthermore, the idea of a body-transfer smart home, which combines a variety of robotic service systems to aid the occupants and a human-friendly user interface to effectively follow the user commands, has been developed and tested. The combination of the body-transfer bed, body-transfer wheelchair, and body-transfer toilet wheelchair enables a patient to be transferred smoothly and securely. All modules on the body-transfer device assist patients in being moved in a supine position

without disturbing a person's posture. Our proposed approach facilitates the patient and completes the movement from bed to wheelchair and vice versa in less than two minutes. Furthermore, this prototype also demonstrates the effectiveness of the BTTW in both toilet and bathroom scenarios. Employing such a system in homes, hospitals, and nursing homes increases the independence of such people, which is the ultimate goal of active assistive living in a smart home. It also mitigates the risk of injuries, burden, pain, and dependence on caregivers. Subsequently, the quality of life, satisfaction, and self-esteem of the person cared for will also increase.

The sliding transfer approach ensures a minimal probability of deformation of the sagittal plane of a person while transferring from a BTB to a wheelchair and vice versa. Contrary to previously developed systems, the BTTW has no electrical devices installed because moisture on the bathroom or toilet floor presents a risk of electrocution of elderly/healthcare professionals. Thus, the BTS safeguards the safety of such a person in bed and bathroom. An intelligent and user-friendly handheld pendant has been developed to manage all the body-transfer modules.

The experience gained from developing a body-transfer system will aid in creating a more human-centered technology that can assist more functionally disabled people. Several IO methods based on facial expression, eye gaze, and bio-signals can be used to deal with various physically impaired people [44]. Additionally, this study had some limitations, including a small sample size and using a prototype in a controlled environment. As a result, these focus group survey findings may not accurately reflect the target audience. Future work on the BTS will include developing a fully autonomous transfer system and testing it in the home and hospital setting to achieve the goal of active assistive living.

**Author Contributions:** Conceptualization, C.-Y.L.; methodology, S.M.; software, B. and H.B.; validation, S.M.; formal analysis, S.M., H.B. and B.; investigation, S.M.; resources, C.-Y.L.; data curation, S.M.; writing—original draft preparation, S.M., H.B. and B.; writing—review and editing, C.-Y.L.; supervision, C.-Y.L. project administration, C.-Y.L.; funding acquisition, C.-Y.L. All authors have read and agreed to the published version of the manuscript.

**Funding:** This work is financially supported by Taiwan Building Technology Center and the Center for Cyber-Physical System Innovation from the Featured Areas Research Center Program within the framework of the Higher Education Sprout Project by the Ministry of Education (MOE) in Taiwan. This research is also financially supported by the Ministry of Science and Technology, Taiwan (R.O.C), Under Grant MOST 108-2221-E-011-140-MY2.

**Data Availability Statement:** The data presented in this study are available on request from the corresponding author. The data are not publicly available due to privacy.

**Conflicts of Interest:** The authors declare no conflict of interest.

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
