# Peer review of "The Design and User Evaluation of Body-Transfer System via Sliding Transfer Approach for Assisting Functionally Impaired People"

_machines, doi:10.3390/machines11050555_

Round 1

Reviewer 1 Report

Critique

This is an extremely interesting and timely manuscript. The issue of transfers in elderly, chronically ill, and disabled populations vitally important for managing care in these populations. Injury from transfers to formal and informal caregivers is an important challenge. Thus, the manuscript's topic is highly significant to home health care and other care environments. While the motivation is highly relevant, this reviewer found the proposed solution to be cumbersome and not likely to be adopted iin a normal care environment (i.e. nursing home, home health care, etc), thereby limiting its use. It would seem a major criteria for such a device, namely safety, is not really considered.  The results shown in Fig 15 seem to indicate lack of enthusiasm and support for this device (more later) and should be re-assessed by the authors (I think they misinterpret their results shown).

Specific comments include:

·         Introduction is well done with clear justification of the need and contributions of this work. This reviewer recommends rewording “major contributions” to “main objectives of this proposed methodology”.

·         Overall, the literature review in section 2 is fine, however the discussion on “smart homes” seems irrelevant and tangential to the primary effort in the manuscript. Suggest removing.

·         The design concept in Section 3 is clearly explained although some design considerations seem lack consideration of how the system will be used in practice. The need for a 3-piece system for transfer detracts from its usability in real-world situations as in the home, assisted living, nursing home environments, there are likely space/storage limitations. 

o   A roller-based system may present safety issues when combined with linens, bedding, blankets and other items found in a bed.

o   Since the intention is to be used for seniors and those with limited mobility, there seems to be lack of consideration for comfort and, more dangerously, the potential for the system to increase the likeliness of developing pressure ulcers/bed sores.

o   The 2-conveyor (upper torso, buttocks area) seems to lack consideration for articulation of the legs from one surface to the next. This is exhibited in Fig 12d where the test subject is shown with legs misaligned.  This misalignment may result in imbalance of the wheelchair(BTW, BTTW) and/or perhaps even injury during transfer.

o   In Section 3.1.3, several equations are given. Some variables are not well explained or shown in Fig 3 (ex. l_abr).  Also, please check equations 1 and 5. Should the masses include the body masses for these moment balances (not just bed component masses)? The center of gravity of the body components may (likely) not be the same as that for the bed components.

o   In section 3.1.4, it would seem an analysis of tipping/stability (i.e. Fig 5b) is warranted due to safety concerns.

o   In Section 3.1.6, line 398 states “as shown in Figure 7(b).” Should this refer to Figure 7(c)?

o   Section 3.1.7 mentions a need for “docking” the wheelchair to the bed.  What does this mean? Is there a locking mechanism that attaches the 2 systems together to prevent movement and/or ensure alignment of the conveyor belts? This does not seem to be the case from Fig 8.

o   The BTTW (bathroom transfer chair) as shown in Fig 13 looks like a series of rollers. It doesn’t not seem to be very comfortable for the intended users. Also, Fig 11 does not show any rollers/conveyor below the buttocks for this system. If not, how does the person transfer from bed to BTTW? Image for BTTW in Fig 11 looks very different than what is shown in Fig 13 (no toilet access). Is there a possible safety issue (falling through with small seniors) with the system shown in Fig 11? This BTTW system seems under developed.

·         Section 4.0 discusses user evaluation and human subjects involvement.  Was this study IRB approved for the protection of human subjects? There is no mention of this and is potentially an ethical concern. There is discussion that subjects “had given informed consent” so perhaps this was missed in the manuscript preparation.

·         Figure 15 should identify which bar correlates to users (left?) vs. caregivers (right?).

o   Fig 15 does not seem to be overwhelmingly supportive of the proposed device (this reviewer does not consider neutral as supportive). Q4 seems to indicate some concerns over safety.

o   Caregivers seem to think that the system is not much benefit (Q6). Discussion of caregiver responses in lines 539-548 do not seem to use data from Fig 15.

Author Response

To begin with, we would like to thank the reviewer for expressing their thoughts and carefully reading our manuscript. We are also thankful for their insightful comments. We have revised the manuscript by adhering to the reviewer's recommendations as closely as possible. We have endeavored to provide all appropriate responses to the questions, to the best of our knowledge.

Reviewer 2 Report

1)     I suggest language used be consistent with ICF guidelines. “Physical disability” is not part of the current nomenclature. “Mobility limitation is probably a better term.  The term paralysis (which is an impairment) doesn’t fit as a synonym with disability either.

2)     I  think the authors understate the number and value of available devices for aiding transfers (eg, lifts, boards, track systems, pivot disks, and low friction fabrics.) *

3)     How much does all this cost?

4)     Passive voice (eg, It has been demonstrated…”) should be avoided.

5)     A statement such as “the majority of the elderly end up in a wheelchair” is difficult for me to believe. A concrete reference supporting such a claim is required.  

6)     References are presented inconsistently.

7)     IRB approval.

* J Rehabil Res Dev 2001;38:135-139

* Arch Phys Med Rehabil 1999;80:851-853

Author Response

(The authors gave the same response as above.)

Reviewer 3 Report

Complete review: The Design and User-Evaluation of Body Transfer System via Sliding Transfer Approach for Assisting Functionally Impaired People.

1. The paper deals assistive devices that help disabled people. In this research, a body-transfer system is designed and developed to ensure that the posture and body angle of the person in the sagittal plane remains unaltered while transferring from bed to wheelchair and vice versa. Two independently controlled conveyor belts mounted on the indigenously developed bed, are employed to transfer the disabled person using a sliding approach. Additionally, a wheelchair with conveyor belts that are fully automated has been used to carry and transfer the user to and from the wheelchair. Furthermore, an integrated control architecture has been developed for safely operating the entire body-transfer system. An experimental assessment of the body-transfer system's performance has been conducted.

2.The mechanical architecture of the entire system demonstrates the effort of the authors to create an original device. Un body-transfer system is designed and developed to ensure that the posture and body angle of the person in the sagittal plane remains unaltered while transferring from bed to wheelchair and vice versa.

3. In this paper, the authors do not demonstrate a high scientific level, but the proposed technical, mechanical solutions correspond to the quality standard for publication in the Journal.

4. The authors carry out a literature investigation of similar techniques and correctly present the advantages of the proposed technical solution.

5. The graphic presentation and the experimental data illustrate well the results of the paper.

6. The Conclusions section summarizes the results of the paper and some future research directions are formulated.

7. I conclude that the paper deals with an interesting and current topic, it is well written and organized and I propose acceptance for publication.

Author Response

(The authors gave the same response as above.)

Reviewer 4 Report

This proposed design is for transferring a patient body from A place to B place. The necessity is absolute, however I cannot agree with the good effect of proposed design on the transfer system. 

Major

1. How can the conveyor belt interlock when moving from A place to B place? 

2. I think that the bigger torque is necessary at the center of mass for the patient body, thus why not operate with the same torque?

3. Hove you considered the patient's skin damage?

4. This experiment must be approved by ethical review. 

Minor

1. The construction of this manuscript is not good. 

2. The related works is so old.

3. The results in figure 4 is strange, not exact. 

Author Response

(The authors gave the same response as above.)

Round 2

Reviewer 1 Report

Thank you to the authors for addressing this reviewer's concerns in the original manuscript in their resubmission. Most concerns have been addressed.  This reviewer still has concerns regarding the safety of the system as well as usability by the intended audience. It seems the charts provided in Fig 17 are for a "presentation" of the BTW system rather than actual usability which limits usefulness. 

The authors mention in their responses that IRB approval has been incorporated into the revision. However, this reviewer did not find this in the revised manuscript. Perhaps the editor can confirm before final publication. 

Overall the revision addressed most of the major technical concerns. Please  provide careful editorial review. Ln 159 should be Figure 1(a). Line 198 has "body-transfer toilet wheelchair" twice in a row. Ln 201, 251, 405 (and elsewhere) has "Safety" or "Transfer" capitalized unnecessarily. Perhaps ln 204 should read: "Two centered AC servomotors mounted opposite each other drive these conveyor belts." Ln 268 refers to F_act whereas eqn 2 refers to F_br. Please standardize subscripts. Please review Eqn 13 as it seems deceleration term is included twice. Review Ln 399-400 as there is a sentence fragment without complete thought ("The force acting on the point N_A and N_B"). Fig 7 refers to "upright" position in Fig 7(b). The position is not upright, but rather horizontal. Perhaps this should read "normal" or "resting" position. There is some redundancy in the last 2 paragraphs in the conclusion (ln 679-681 and ln 682-684).

Author Response

To begin with, we would like to thank the reviewer for expressing their thoughts and carefully reading our manuscript. We are also thankful for their insightful comments. We have revised the manuscript by adhering to the reviewer's recommendations as closely as possible. We have endeavoured to provide all appropriate responses to the questions, to the best of our knowledge.

Reviewer 4 Report

I think that you are revising the manuscript.

Please check your submitted file.

It looks not ready for the review processing. 

Then, I have no idea which point was changed according to my previous comments.

Author Response

(The authors gave the same response as above.)
